# Factors shaping network emergence: A cross-country comparison of quality of care networks in Bangladesh, Ethiopia, Malawi, and Uganda

**Yusra Ribhi Shawar** [1,2]*, **Nehla Djellouli** [3], **Kohenour Akter** [4], **Will Payne** [1], **Mary Kinney** [5], **Kasonde Mwaba** [3], **Gloria Seruwagi** [6], **Mike English** [7], **Tanya Marchant** [8], **The QCN Evaluation Group** [¶], **Jeremy Shiffman** [1,2], **Tim Colbourn** [3]

**1** Department of International Health, Bloomberg School of Public Health, John Hopkins University, Baltimore, Maryland, United States of America, **2** School of Advanced International Studies, John Hopkins University, Washington, District of Columbia, United States of America, **3** Institute for Global Health, University College London, London, United Kingdom, **4** Perinatal Care Project, Diabetic Association of Bangladesh, Dhaka, Bangladesh, **5** School of Public Health, University of the Western Cape, Cape Town, South Africa, **6** School of Public Health, Makerere University, Kampala, Uganda, **7** Centre for Tropical Medicine and Global Health, University of Oxford, Oxford, United Kingdom, **8** Department of Disease Control, London School of Hygiene & Tropical Medicine, London, United Kingdom

¶ Membership of the QCN Evaluation Group is listed in the Acknowledgments.
* yusra.shawar@jhu.edu

**Data Availability Statement:** All data is derived from qualitative interviews, most with stakeholders where only one individual holds a position, either

## Abstract

The Quality-of-Care Network (QCN) was conceptualized by the World Health Organization (WHO) and other global partners to facilitate learning on and improve quality of care for maternal and newborn health within and across low and middle-income countries. However, there was significant variance in the speed and extent to which QCN formed in the involved countries. This paper investigates the factors that shaped QCN's differential emergence in Bangladesh, Ethiopia, Malawi, and Uganda. Drawing on network scholarship, we conducted a replicated case study of the four country cases and triangulated several sources of data, including a document review, observations of national-level and district level meetings, and key informant interviews in each country and at the global level. Thematic coding was performed in NVivo 12. We find that QCN emerged most quickly and robustly in Bangladesh, followed by Ethiopia, then Uganda, and slowest and with least institutionalization in Malawi. Factors connected to the policy environment and network features explained variance in network emergence. With respect to the policy environment, pre-existing resources and initiatives dedicated to maternal and newborn health and quality improvement, strong data and health system capacity, and national commitment to advancing on synergistic goals were crucial drivers to QCN's emergence. With respect to the features of the network itself, the embedding of QCN leadership in powerful agencies with pre-existing coordination structures and trusting relationships with key stakeholders, inclusive network membership, and effective individual national and local leadership were also crucial in explaining QCN's speed and quality of emergence across countries. Studying QCN emergence provides critical insights as to why well-intentioned top-down global health networks may not materialize

within federal or state government, facilities, or NGOs. Every care has been taken to ensure anonymity of the data in the submitted manuscript but the authors from all 4 countries feel strongly that making data freely available would jeopardise the conditions of informed consent. We have uploaded a detailed methods document within the Supporting Information files. Data requests may be sent to the UCL data protection office: data-protection@ucl.ac.uk and UCL ethics committee ethics@ucl.ac.uk.

**Funding:** This work was funded by the Medical Research Council (MRC) Health System Research Initiative 5th call via grant MR/S013466/1 to TC at UCL Institute for Global Health, United Kingdom, YRS and JS at Johns Hopkins University, United States of America, KA and AK at Diabetic Association of Bangladesh Perinatal Care Project, Bangladesh, CM at Parent and Child Health Initiative, Malawi, GS at Makerere University School of Public Health, Uganda, and ME at University of Oxford, United Kingdom; and by the Bill & Melinda Gates foundation via grant INV-007644 to TM at LSHTM, United Kingdom. The funders had no role in study design, data collection and analysis, decision to publish, or preparation of the manuscript.

**Competing interests:** The authors have declared that no competing interests exist.

in some country contexts and have relatively quick uptake in others, and has implications for a network's perceived legitimacy and ultimate effectiveness in producing stated objectives.

## Introduction

Proliferating in global health over the last three decades, networks are a growing subject of attention given their prevalence, critical role, and impact on population health outcomes [1–8]. Global health networks—"cross-national webs of individuals and organizations linked by a shared concern to address a particular health problem global in scope" [1]— take on a wide variety of forms and purposes, and are often engaged in research, advocacy, and/or program or policy design, learning, implementation or evaluation [7]. They often engage a wide variety of actors including academic institutions, governments, international organizations, UN agencies, foundations, and service providers [1]. Some networks focus on policy consequences and public goods development and provision—known as global public policy networks [9]. Others focus on knowledge generation and identification of causal relationships—known as epistemic communities [10, 11]. And other networks are defined by their principled ideas and advocacy —known as transnational advocacy networks [12]. Networks may be formal in nature, such as the former Partnership to End Violence Against Children [13]; or more informal and dynamic such as the largely low and middle-income country (LMIC)-based network of physicians concerned with rheumatic heart disease [14].

Despite growing scholarship on the subject [15–18], relatively little remains known about how global health networks form and evolve. Specifically, much less research has examined the emergence of global health *implementation* networks within country settings. Why do formal global health networks—largely initiated and conceptualized among international organizations and partners to improve health outcomes—form more easily within some countries and struggle to crystalize in others? We explore this question through an examination of the Network for Improving Quality of Care for Maternal Newborn and Child Health, also known as the Quality-of-Care Network (QCN) [19].

Formally launched in 2017, QCN was conceptualized by the World Health Organization (WHO) and other global partners to facilitate learning and improvement of quality of care within and across LMICs in order to reduce mortality risks for millions of women and newborns and make progress towards achieving the Sustainable Development Goals (SDGs) and universal health coverage [20]. QCN seeks to bring together historically siloed maternal, newborn, and child health (MNCH) interventions and broader quality improvement efforts [21, 22]. All eleven 'pathfinder' LMICs that joined QCN exhibited a high maternal mortality rate, and their governments made some level of political commitment to improving quality of care for maternal and newborn health. Despite these similarities, there was a clear spectrum of QCN emergence across the pathfinder countries. This paper investigates the factors shaping variance in the QCN's emergence in four of the pathfinder countries: Bangladesh, Ethiopia, Malawi, and Uganda. We chose these four countries because they represent a range of contexts and starting points.

Studying QCN *emergence* drawing on social science scholarship [23–29]—provides critical insights as to why some well-intentioned top-down global health networks do not materialize in some country contexts and have relatively quick uptake in others. The pattern, speed, and extent of a network's initial emergence is likely to reflect and have implications on a network's legitimacy, configurations and interactions among involved actors, effectiveness in producing

stated objectives, and sustainability—each of which are discussed respectively by Akter et al, [30] Mukinda et al, [31] Tesfa et al, [32] Mwandira et al, [33] Djellouli et al, [34] and Lemma et al, [35] in this paper series concerned with QCN [S1 Text]. An understanding of the key factors shaping variable network emergence across these countries is critical to WHO and global partners, as they seek to establish QCN activities in other countries, but also more broadly for policymakers seeking to initiate global health networks and partnerships across various contexts.

## QCN emergence at the global level

The idea of QCN arose from pre-existing efforts to improve global MNCH and the increasing emphasis on quality of care among global actors. From 2014 to 2016, UNICEF's Every Mother Every Newborn (EMEN) project, which was funded by the Bill and Melinda Gates Foundation (BMGF), sought to establish mother and baby friendly hospitals across LMICs [36]. In August 2016, WHO published Standards for Improving Quality of Maternal and Newborn Care in Health Facilities [37]. Thereafter, individuals at the WHO, the Institute for Healthcare Improvement (IHI), and UNICEF began to discuss the possibility of developing an implementation strategy for these standards, motivated to see the standards translated into sustained results. WHO then gathered several multi-sector partners, including donors, academics, governing bodies, and country technical partners, such as USAID, Jhpiego, UNFPA, and others, to discuss what this could look like. These initial talks raised awareness of the problem that there were many actors engaged in MNCH and quality of care in various countries but that each was working in silos, with few examples of successful, consistent, scalable, and sustainable approaches to the many facets of improving quality of care for mothers and children. This observation led to the idea of a network where all these actors could attempt to work together and both partners and countries could share and learn from one another to establish one joint approach. Following initial conversations, WHO's director for the Department of Maternal Newborn Child Adolescent Health and Ageing advanced the idea of establishing a learning network across LMICs with the primary aim of improving maternal and neonatal survival.

In October 2016, WHO and partners approached nine pathfinder countries (Bangladesh, Côte d'Ivoire, Ethiopia, Ghana, India, Malawi, Nigeria, Uganda and the United Republic of Tanzania) to join the network; Sierra Leone joined in 2017 and Kenya in 2019. These countries were asked to join the network because they were perceived by WHO and partners to be well-positioned to make rapid progress: each country demonstrated political will and commitment for improving MNCH, as well as strong funding and technical support from partners [38]. From the outset, QCN aimed to build on ongoing efforts rather than establish an additional silo. QCN differed from many prior global health networks in that its purpose was to operationalize quality improvement within countries to reduce mortality, rather than draw donors' and global actors' attention to specific health challenges [5, 8, 13, 14, 39].

QCN was officially launched in Lilongwe, Malawi in February 2017 [40]. Strategic objectives were announced and an official guide to the network's goals and intentions was distributed [41] to the 340 representatives of the ten initial pathfinder countries and global partners in attendance. The partners produced an agreement on the primary network aims: 1) reduce maternal and newborn mortality, and specifically reduce maternal and newborn deaths and stillbirths in participating health facilities by 50% over five years; and 2) improve the experience of care [20, 37]. In order to accomplish these goals, it was agreed to apply the WHO's "QED" (Quality, Equity and Dignity) and "LALA" (Leadership, Accountability, Learning, and Action) implementation frameworks to quality of care initiatives in partner countries [39, 42], Subsequent international meetings facilitated global QCN emergence and development,

including meetings in Dar es Salaam, Tanzania in December 2017, and in Addis Ababa, Ethiopia in March 2019, which involved 11 additional observer countries [43]. As noted in WHO's 2021 QCN progress report [44] and further discussed in the QCN meeting in Ghana in March 2023, the expectation was that resources—technical and monetary—for supporting each country network and their broader quality of care improvement initiatives were to be transitioned from global to domestic actors. However, few countries achieved this during the network' first five years, except for Bangladesh and Ethiopia. Fig 1 details the timeline of the emergence of QCN at the global level, and when countries joined.

In terms of the network's structure at the global level: WHO was the global coordinating body; it provided guidance on how to improve the technical quality of care, while also advising on monitoring and evaluation, and organizing formal multi-country, multi-stakeholder network engagement, which facilitated learning, especially between the countries. Other partners included other UN agencies, such as UNICEF, co-leading QCN with WHO, and UNFPA, providing technical support; BMGF, providing funding to the QCN Secretariat and to UNICEF in-country for national implementation; IHI, leading the development of quality improvement approaches in-country and providing technical support; and USAID, a key implementing partner in-country. The engagement of high-level staff from UN agencies and prominent funders were critical to the strong emergence of QCN at the global level [45].

## Methods

### Theory: Global health network emergence

This analysis draws on scholarship from various disciplines [24, 25] that examine networks broadly, but also global health networks and global health partnerships specifically. Network "emergence" is defined as the crystallization of an organized group of actors around a shared problem [1]. We understand network emergence to be on a spectrum, rather than binary [26–28], exemplified by: 1) radical novelty (i.e. displaying a new property or function); 2) coherence (i.e., maintenance or consistency over a period of time); 3) a global or macro level (i.e., the entirety being interconnected); 4) ostensibility (i.e., it can be perceived or the member parts are aware of it) [29]. In this paper, we characterize network emergence in each country in terms of the quality of communication and coordination of members across various levels, the level of institutionalization of network's goals and activities in country structures, as well as the speed of formation.

Several factors likely shape network emergence [1, 2, 46–50]. We modified these for relevance to global health network emergence within country settings and group them into two categories of factors: policy environment and network features. Table 1 presents the factors shaping global health network emergence according to these two broad categories. Policy environment pertains to developments and dynamics external to the network. Specifically, these encompass the pre-existing structures and dynamics that emerging networks must navigate. According to social science scholarship [1], a network is more likely to strongly emerge—that is, crystalize more quickly and be of greater quality—in-country when: there are established country policies, programs, and funding dedicated to the issue of concern; there are country systems that are capable of collecting and reporting on the severity of the problem; and the country's political elites—as well as the frontline workers—are motivated to advance the issue and believe they are able to address it.

A second set of factors concerns network features, which involve the strategy, structure, and attributes of the actors that constitute the network. Specifically, scholarship on collective action points to two network features that may be particularly influential for network emergence: its governance and leadership [1, 2]. Concerning governance, a network is more likely

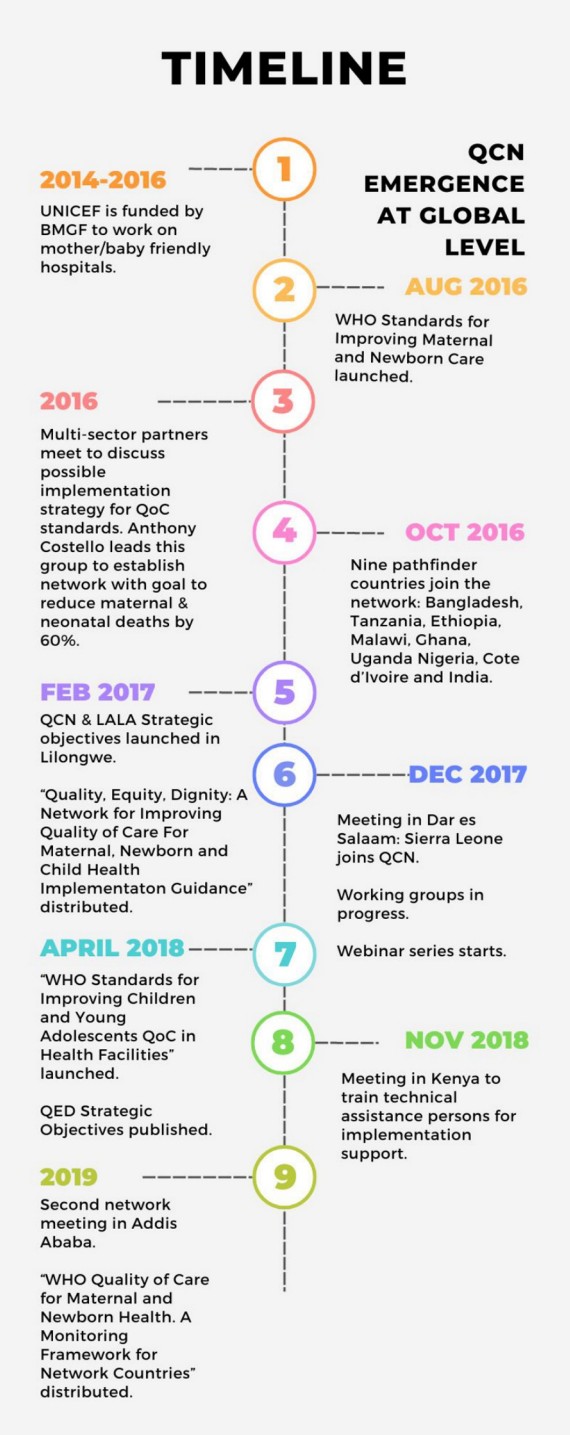

**Fig 1. QCN emergence on global level timeline.**

to strongly emerge in a country when it includes locals in decision-making processes, network members are embedded in powerful agencies or hold key decision-making positions, and there are pre-existing and trusting relationships between involved actors; this is likely to contribute to effective communication and coordination among involved actors in achieving

**Table 1. Factors shaping global health network emergence and effectiveness.**

| | Impact on Network Emergence |
|---|---|
| **Policy Environment** | |
| **Resources and initiatives dedicated to the issue in-country** | Countries with existing policies, programs, and funding for an issue will more likely facilitate network emergence for that issue, given that these resources reflect pre-existing commitments and actors dedicated to the issue. In contrast, a dearth of such resources is likely to hinder motivation for new efforts, as the creation of the network is more likely to be seen as intractable or unsustainable. |
| **Data and health system capacity in-country** | Systems that are capable of collecting and reporting on the severity of the problem, as well as those that have health systems that are in a better position to take up relevant initiatives (i.e., quality improvement), will lead to network emergence, given that there is a basis for coming together given demonstrated burden and demonstrated tractability. |
| **Political developments and legacies/country leadership priorities in-country** | Networks are more likely to emerge when political elites have interest in advancing the issue, and there are 'policy windows' that highlight their need; also, networks are more likely to emerge in politically stable environments (no domestic conflict) given that network establishment requires movement and coordination, which are likely to be obstructed by environments with domestic conflict and/or tumultuous political leadership. |
| **Nature of Network** | |
| **Governance** | Networks are more likely to emerge when they include national and local stakeholders in decision-making processes, and some of these stakeholders are embedded in powerful agencies or hold key decision-making positions, and there are pre-existing and trusting relationships between involved actors at the international, national and sub-national levels. These pre-existing interactions, and the trust they create among involved actors, are foundational for generating networked activities. |
| **Leadership** | Networks are more likely to emerge if effective leaders exist, capable of bringing relevant actors across a network together, and once linked, in guiding them to effective collective action for establishing a network. |

collective goals. Concerning leadership, networks are more likely to emerge when there are long-standing and respected in-country champions dedicated to the subject, capable of directing and sustaining the network's growth and development [51, 52].

## Replicated case study and case justification

We conducted a replicated case study [53]—a method relying chiefly on within and between-case analyses. We employed a most similar case approach given the four country cases—Bangladesh, Ethiopia, Malawi and Uganda—are similar in a number of ways (i.e., issue characteristics: the QCN network across each of these countries seeks to improve MNCH outcomes) but vary on a number of variables of interest (which in this case, are network features and policy environment). We utilized a process tracing methodology [54], a qualitative case study research strategy commonly employed in political science that seeks to uncover the mechanisms that underpin cause and effect relationships. In doing so, we explored why the causal mechanisms (i.e. in this case, relying solely on issue characteristics in explaining network emergence) break down and what other factors must be present to explain variance more appropriately in network emergence across the four cases. The objective is explanatory: to probe for new—but yet unspecified—explanations and identify alternative causal factors.

## Data collected

We triangulated across several sources of data including a document review, observations, and key informant interviews. With respect to the document review, we reviewed all accessible published and unpublished documents and communications relating to the QCN at global level and at national and sub-national levels across the four countries of study. These included strategy and management documents, operational plans, directives, formal minutes, and reports.

In the supplement, we summarize the document review pertaining to each country and at the global level [S2 Text.]. We were able to access unpublished documents via WHO, as well as QCN contacts within the Ministries of Health in the four countries of study. In terms of observations, we conducted non-participant observations [55] of multi-country meetings and key national-level and district-level meetings. Activities at district level were also observed via visits to two better- and two least-performing QCN hospitals in several iterative rounds. This is summarized in Table 3 of the S2 Text. Best- and worst- performing sites were selected based on maternal and newborn health outcomes and other quality of care data (e.g., those used in national schemes). We used templates to capture key processes relevant to the focus of the network at each site during observations, as well as unstructured notes. The observations were used, among other purposes, to explore whether actors involved at various levels feel anything has changed over the period of QCN operation. To ensure our observations were informative, they were conducted by trained and experienced researchers familiar with the local setting and recorded in detailed field notes.

We also conducted semi-structured interviews with global, national, and local level network members and key stakeholders, including employees of implementing INGOs, academic partners, representatives of the countries' Ministries of Health, and managing actors and clinicians within district hospitals and health sites. Details of how these were conducted are provided in the supplementary material [S2 Text]. We sought to pay particular attention to the perspectives and goals of those carrying out the work of the network [56, 57]. At the national and local levels, we conducted several iterative rounds of interviews across the four countries, at least six months apart, to capture changes in how the network was operating and views pertaining to network activities as well as follow-up on emerging findings from the previous round. The number of rounds and the number of interviews with global, national, and local level stakeholders in each round are provided in Table 1 of the S2 Text.

## Data analysis

The analysis was iterative, exploring emerging findings and drawing on network and global health partnership scholarship and frameworks [58], resulting in use of both inductive and deductive approaches [55]. While our qualitative codebook remained closely grounded in social science scholarship concerning network emergence, we modified key informant selection and the content of our interview guides throughout the study to invite participants to reflect on earlier findings, and to address topical blind spots. Coding was performed in NVivo 12, using a codebook consisting of "case study" codes and "theory" codes. The bulk of our coding framework and qualitative data analysis for interview, observation, and document review data consisted of a deductive, manifest content analysis process. In this process, predetermined codes, largely based on theoretical frameworks referenced in the theory section [1, 59] were applied to meaning units—the smallest unit of text which answered the research objective—in each file [60].

Our codebook was prepared via generation of an outline of each relevant theory (summarized in S2 Text). The codebook provided to the project's coding staff included reference

papers for further reading, as well as detailed descriptions of each theoretical unit and of each code. Coders were also introduced to the codebook and coding software via a global webinar, and coders underwent an iterative training and standardization processes in which groups of coders, including at least one project lead researcher having close familiarity with the project theoretical components, identified and coded meaning units in a closed set of 3–5 files, and results were compared and discussed [61]. This allowed us to initiate a relatively high level of inter-coder consistency.

We applied a process of decontextualization, recontextualization, categorization, and compilation in conducting our qualitative data analysis [60]. In decontextualization [62], each file was read by the coder in order to understand its context and content, and meaning units were identified and labeled with a code. Immediately subsequent to decontextualization, the coder reviewed the same document again, observing areas in which no codes had been applied, and determining whether these areas were relevant to the research aim. Additional codes were then applied if the content was determined to be relevant [60, 63]. The categorization process was relatively straightforward due to our use of a deductive codebook, as analysis was conducted by viewing the meaning units regrouped to each theoretical component, including the relevant headings and sub-headings suggested by the theory [60]. Finally, our compilation process involved manifest analysis, via the preparation of narrative qualitative syntheses that employed both direct quotes and narrative summaries of the direct textual content of each file [60]. These documents took the form of informal working papers prepared on each theoretical topic under analysis and offered an additional opportunity for "sanity checking" our conclusions with broader groups of in-country and global researchers who were not directly involved with the coding and analysis process. Due to the lengthy and iterative nature of our data collection process, the preparation of these interim syntheses also presented an opportunity to reformulate the qualitative interview guide when information was determined to be missing or incomplete.

## Ethics

Ethical approval was received from University College London Research Ethics Committee (ref: 3433/003); London School of Hygiene and Tropical Medicine (17541); BADAS Ethical Review Committee (ref: BADAS-ERC/EC/19/00274), Ethiopian Public Health Institute Institutional Review Board (ref: EPHI-IRB-240-2020), National Health Sciences Research Committee in Malawi (ref: 19/03/2264) and Makerere University Institutional Review Board (ref: Protocol 869).

## Results

### QCN emergence at the national level: Bangladesh, Ethiopia, Malawi, and Uganda

In tandem with the emergence of QCN's structure at the global level, in-country networks emerged in Bangladesh, Ethiopia, Malawi, and Uganda between 2016 and 2019. Despite involvement of many of the same actors across the four countries (summarized in S1 Table), the quality and speed of QCN's emergence varied (summarized in Table 2). In Bangladesh, and to some extent Ethiopia, QCN structures were established more quickly, with in-country institutionalization of a global network's goals and activities, and communication and coordination of actors across the global, national, and local levels. In contrast, in Malawi and Uganda, QCN emergence occurred at a slower pace and there was fragmentation—and in some cases, a complete lack of awareness—of activities and among involved actors.

**Table 2. Emergence of QCN in four pathfinding countries.**

| | Bangladesh | Ethiopia | Uganda | Malawi |
|---|---|---|---|---|
| **Overall strength of QCN emergence** | **Strongest** | **Strong** | **Moderate** | **Weak** |
| **Speed of initiation** | **Fastest:** 2016 | **Fast:** 2017 | **Slow:** 2017; however, roadmap and operational plan developed in 2019 & 2020. | **Slowest**: Early to mid-2019. |
| **QCN institutionalization of the proposed goals and activities into local structures** | **Strong**: QCN integrated within Bangladesh's Quality Improvement Secretariat. | **Strong/Moderate**: The new quality directorate supported QCN institutionalization; however, some key actors initially not involved | **Moderate/Weak**: Difficulties finding appropriate institutional host placement. | **Weak**: Some actors unaware they are part of QCN |
| **QCN coordination across relevant global, national, and local actors** | **Moderate**: The goals among global and national actors aligned, and mechanisms in place to foster communication and coordination to implement QCN activities. However, coordination with local level lagging. | **Moderate**: Communication and coordination with local level lagging. | **Weak**: Initially no implementation or monitoring plan, and fragmentated implementation of activities. | **Weak**: Initially no/little communication with learning districts. |
| **Rationale for country OCN engagement** | Joint effort by Government and WHO; country members perceived to be pioneering work in quality improvement. | Introduced in meeting called by WHO. | Initiated by WHO because of low performance in maternal and newborn healthcare and existing quality work. | Initiated by WHO because of low performance in maternal and newborn healthcare. |

**Emergence of QCN in Bangladesh.** In Bangladesh, many respondents described their country's engagement with QCN to be a result of their pioneering efforts in quality improvement. Several in-country respondents perceived QCN's emergence as a joint effort by the government and WHO. An implementing partner in Bangladesh described the network's emergence:

> "Both WHO and Government [took the] initiative [together] because if the government does not want, it will not be done. If WHO does not want, it will not be done as well. As both have agreed, it has been done."

Some connected the country's joining QCN to UNICEF's Every Mother, Every Newborn (EMEN) project (2016–2018), and specifically the attendance of the UNICEF Bangladesh country office at the global meeting in 2016. After expressing interest in joining the network, it was discussed with the government, which initiated its establishment in the country. QCN was integrated within Bangladesh's Quality Improvement Secretariat (QIS)—which had been at the center of the country's quality improvement activities since 2015. Housed within the Health Economics Unit (HEU), QIS serves as a formal management body of the National Quality Improvement Committee. Other long-standing development partners within the country were equally engaged as part of the network at the highest levels, including WHO, UNICEF, UNFPA, Save the Children, and USAID. Various professional associations, such as the Obstetrical and Gynaecological Society of Bangladesh, and global partners, such as University Research Co. (URC), were involved to provide technical support and capacity building. The goals among global, national, and to some extent local actors aligned and mechanisms were put in place relatively quickly to foster communication and coordination across these actors—from the global QCN Secretariat to the local Civil Surgeons—to implement QCN activities. Beginning in 2016—even before the formal launch of QCN—the Ministry of Health and Family Welfare had already identified and began work in the first round of learning districts—Kurigram and Narsingdhi—as part of QCN [64]. Rapid scale up to additional districts followed shortly after.

**Emergence of QCN in Ethiopia.**    In Ethiopia, unlike Bangladesh, respondents reported WHO as fully initiating the emergence of the national network by inviting their country to be part of QCN in a meeting in 2016. One Ethiopian national partner described the relatively weaker communication between the Ministry of Health and its partners during QCN's establishment:

> "In 2016, we went to Geneva to evaluate the work of [maternal and newborn health]. At the meeting, I heard that QCN is to be launched by the initiation of WHO. At the time we heard that the document is prepared and countries were selected. . .Prior to the meeting, WHO had discussed with the government and Ethiopia was already selected."

In 2017, the Ministry of Health in collaboration with WHO further discussed and subsequently established the network in Ethiopia. Led by the WHO and the Ministry of Health, they identified interested partners, created forums and mobilized resources. QCN members were initially coordinated by the Ministry of Health with substantial support from WHO; together they set the goals, developed a roadmap and guidelines, and facilitated the implementation of the network in the selected facilities, with the support of implementing partners—such as IHI, Clinton Health Access Initiative (CHAI), WHO, Transform HDR, and Transform PHCU. The network emerged relatively fast and coordination, recognition, commitment and supervision of QCN was initially strong at the national level. The new quality directorate supported its institutionalization, and there was strong alignment with government policies and pre-existing large-scale programs on quality improvement. By 2017, a technical working group (TWG), consisting of representatives of different partners, was established, a national roadmap [17] called LALI (Leadership, Accountability, Learning, Implementation, alternatively used to LALA) was developed, and 48 learning facilities were identified and selected quickly. However, the network initially faced challenges penetrating sub-national levels, where communication and establishment of coordination structures lagged, and many were unaware of the roadmap. Furthermore, unlike Bangladesh, Ethiopian respondents noted that several key actors—including global implementing partners, local institutions, and some key units in the MoH structure that worked on mothers' and children's health—were initially absent from the network, hindering some institutionalization of the network's activities. A sub-national respondent described some of the actors that were missing from the network:

> "Institutions who work in relation to mothers and children's health should have been involved. For example, Save the Children works on child health projects in our region. They should have been members of the MCH unit . . .They are doing the same job and it avoids duplication of resources."

**Emergence in Uganda.**    In Uganda, like Ethiopia, it was perceived that WHO fully initiated QCN. Respondents stressed that their country was participating in QCN because of its low performance in maternal and newborn healthcare. Their country's statistics remained poor despite a significant increase in facility births. A respondent working in the Ministry of Health in Uganda noted how this justified and catalyzed engagement in QCN:

> "Looking at our data, we were seeing that despite having mothers delivering from medical facilities, under skilled health care workers, we were not seeing much more reduction in maternal deaths and newborn deaths. So we felt there was need for us to focus on quality of

the services rather than just pushing more mothers into our facilities. So, the time was right for us to start implementing this."

Like Bangladesh and Ethiopia, an existing structure for improving quality of care was in place in Uganda. However, unlike the former countries, the network's emergence was initially more disjointed: facility-level teams and activities related to quality of care for MNCH were largely dysfunctional, with some quality improvement teams at the facility level not fully operational and little clarity among those involved about the extent of their ongoing activity. While Uganda formally launched QCN in 2017, the roadmap and operational plan were developed later; the former not until 2018 (and later revised in 2020) and the latter not until the end of 2019. The network would only begin to take shape in 2019 with the appointment of a focal person dedicated to QCN and the network's new placement in the renamed Standards Compliance Accreditation Patient Protection (SCAPP) department under the Directorate of Governance and Regulation. In its nascency, the network was characterized as lacking clarity and coordination, with many involved also lacking awareness of the roadmap and operational plan, and no monitoring plan to keep involved actors accountable.

**QCN emergence in Malawi.**   In Malawi, WHO invited the country to join QCN. Similar to Uganda, Malawi was especially eager to join given its poor MNCH indicators. Stakeholders from the Ministry of Health perceived the QCN as an opportunity for Malawi to learn about best practices to reduce maternal mortality and improve the quality of care offered to mothers and children. One implementing partner respondent noted this:

"I think government made a decision that they will join this quality-of-care network so we can improve the quality of care for everybody in Malawi."

QCN was placed in the Government's Quality Management Directorate (QMD), which was a small unit within the Ministry of Health. The WHO assisted the Ministry in gathering key stakeholders, which formed the TWG coordinating body that was charged with planning and the implementation of QCN in the country. In addition to the Reproductive Health Directorate, other key stakeholders at the national level included UNICEF, UNFPA, GIZ, which were essential technical and funding partners that coordinated network efforts and supported other community-based organizations (e.g. Society of Medical Doctors and MaiKhanda) directly to implement quality of care activities. However, QCN activities were not initially institutionalized; the network built few ties with the nursing and clinical departments, as well as the community health service division and district councils—especially crucial actors given the health system's decentralization in Malawi. Other key partners were also absent from QCN in Malawi, including DFID and Save the Children. Similar to Uganda, there was initially little communication and coordination across network actors, especially at the local levels. QCN coordinating meetings were irregular and there were even periods of complete inactivity. Furthermore, learning sites were not selected until mid-2019, and many actors in the learning facilities were initially either unaware that they were learning sites or what their roles and responsibilities were as part of QCN. It was not until this time—almost three years from QCN's official launch in Malawi—that the network began to take shape.

## Factors that explain QCN in-country emergence

Factors connected to the policy environment and network features in Bangladesh, Ethiopia, Malawi and Uganda explain the variance in QCN's emergence across the countries. These are summarized in Table 3.

**Table 3.  Factors that shaped emergence of QCN in four pathfinding countries.**

| | Bangladesh | Ethiopia | Uganda | Malawi |
|---|---|---|---|---|
| **Policy Environment** | | | | |
| **Resources and initiatives dedicated to the issue** | **Strong**<br>• Quality improvement initiatives predate QCN by several decades<br>• Several ongoing, nationwide QI activities | **Strong/Moderate**<br>• Successful national network initiatives<br>• Several nationwide quality improvement activities<br>• However, quality units at lower levels newly established | **Moderate**<br>• Prior quality improvement initiatives not specifically focused on maternal and newborn health | **Weak**<br>• Actors not aware of specifics of past QI initiatives<br>• Lack of national strategy to guide quality improvement efforts |
| **Data and health system capacity** | **Strong**<br>• Many health system capacity challenges concerning human resources and infrastructure<br>• Morale and motivation relatively high among health workforce<br>• Foundational work on national data system | **Weak**<br>• Many health system capacity challenges concerning human resources and infrastructure<br>• Morale and motivation relatively low among health workforce<br>• Foundational work on national data system; however new QCN indicators do not align | **Weak**<br>• Many health system capacity challenges concerning human resources and infrastructure<br>• Morale and motivation relatively low among health workforce<br>• Lack of national monitoring and evaluation framework | **Weak**<br>• Many health system capacity challenges concerning human resources and infrastructure<br>• Morale and motivation relatively low among health workforce<br>• Foundational work on national data system; however new QCN indicators do not align |
| **Political developments and legacies/country leadership priorities** | **Strong**<br>• Strong commitments (i.e., SDGs, universal health coverage, and national health sector plan) | **Weak**<br>• Strong commitments, but:<br>• Low healthcare funding and high out-of-pocket expenditure<br>• Political transition and unrest | **Moderate**<br>• Strong commitments, but:<br>• Low healthcare funding | **Moderate**<br>• Strong commitments, but:<br>• Low healthcare funding and frequent budget cuts |
| **Nature of Network** | | | | |
| **Governance** | **Strong**<br>• Integration in strong government agency: Quality Improvement Secretariat (QIS)—responsible for setting quality standards and introducing QI improvement procedures within medical facilities since 2015<br>• Long history and high level of interaction and trust between many QCN implementing agencies and government actors | **Moderate**<br>• Integration in strong government agency: Federal Ministry of Health of Ethiopia, which had strong QI infrastructure prior to 2015<br>• Some history and level of previous interaction and trust between QCN implementing agencies and government actors<br>• Facility selection lacks adequate representation | **Weak**<br>• Initially, led largely by WHO and partners, with support from Ministry of Health (embedded in relatively newer and weaker government unit)<br>• Implementing partners lack trust and working independently of one another<br>• Lack of community engagement in network | **Weak**<br>• Integration in relatively smaller and weaker unit in Ministry of Health: Quality Management Directorate<br>• Relatively smaller previous history of interactions between QCN implementing partners and government<br>• Majority of involved partners requested to be in QCN vs. being invited or sought out |
| **Leadership** | **Strong**<br>• Long-standing, well-respected and charismatic leaders concerned with quality improvement | **Moderate**<br>• Initially, no focal point for QCN | **Moderate**<br>• Initially, no focal point for QCN and leadership capacity varying across districts | **Weak**<br>• Poor leadership, especially at lower levels |

**Policy environment.**   In terms of policy environment, a country's pre-existing resources and initiatives dedicated to MNCH and quality improvement, data and health system capacity, and politics and leadership priorities were found to shape the emergence of QCN in each of these countries.

*Resources and established country policies, programs, and initiatives dedicated to the problem.* Pre-existing resources, policies, programs, and funding for MNCH and QI helped facilitate QCN emergence. These pre-existing resources fostered and/or were reflective of critical components in existence—such as a highly competent workforce, trusting working relationships with development partners, and perceived synergistic goals—that are crucial for establishing a cohesive network of actors. While pre-existing initiatives for MNCH and quality improvement were apparent in all four countries, their scope and sustainability varied. In Bangladesh, quality

improvement initiatives predated QCN by several decades, fostering a highly competent and effective cadre of national health leaders who understood the potential benefits offered by an international knowledge-sharing network and were willing to devote efforts in seeing through its creation. Respondents working in Bangladesh at Save the Children and UNICEF—the two QCN implementer partners—reported:

> "The Government has many initiatives especially in the context of quality of care. It's basically a government program."

> "Before. . .this network, we have quality improvement program here. So Bangladesh was. . . leading compared to the other countries."

Bangladesh's first quality initiative, 'Quality Assurance Project', was piloted in some hospitals in 1994. Based on the experience of this piloting project, the quality assurance program was included in the Health and Population Sector Program (1993–2003) and continued until 2010. Subsequently, the paradigm shifted to 'Quality Improvement Approach', with attention to quality growing with the adoption of the healthcare financing strategy (2012–2032). At the time that QCN began to emerge in Bangladesh, there were various agencies engaged in several ongoing nationwide quality improvement activities concerning MNCH. These included the Every Mother, Every Newborn (EMEN) project (UNICEF initiated in 2015) [65], the maternal and perinatal death surveillance and review (MPDSR) (WHO, UNICEF, and UNFPA were involved) [66] and the Maternal Newborn Care Strengthening Project (MaMoni MNCSP) (funded by USAID, implemented by Save the Children, and initiated in 2007 and then subsequently again in 2018) [67]. Many respondents noted these and other positive partnerships that existed between the government and development partners prior to QCN. One respondent described how UNICEF supported government-owned facilities according to the framework and guidelines of the WHO, helping them with "capacity building, assessment, minor renovation, their training; [and with] any change [that] is needed [for improving] quality of health services for the mother and children". In Bangladesh, QCN was perceived to build on existing quality improvement practices and existing relationships with development partners, which could consolidate and further positive trends that were already underway.

Similarly, in Ethiopia, quality improvement initiatives by the Ministry of Health, IHI, WHO, and CHAI—all of whom were QCN partners—pre-existed. The government demonstrated strong commitment to improving MNCH between 1990 and 2015 [68, 69]. For example, the 2015 Health Sector Transformation Plan emphasized quality and equity as core pillars [70], the 2016 National Quality Strategy [71] prioritized maternal and newborn health and was co-developed by the Ministry of Health with support from the IHI, and there were successful national networking initiatives such as the Ethiopian Primary Healthcare Alliance for Quality (EPAQ) and Ethiopian Hospitals Alliance for Quality (EHAQ). Several respondents noted how QCN's emergence was facilitated by EHAQ [72]—an initiative established in 2012 by the Ministry of Health to improve the quality of hospitals through collaborative learning. A national-level respondent explained:

> "This network [QCN] was not the first here in Ethiopia; EHAQ was the biggest network countrywide and its experience made us familiar with this one [QCN]. So, this is a favoring opportunity."

Unlike Bangladesh, respondents in Ethiopia reported that the quality units at regional or lower levels were either newly established units or didn't exist at all, despite its specification in

the National Quality Strategy. They also reported challenges with the quality units aligning activities conducted by other units, especially those concerned with maternal and child health (MCH):

> "The other thing is there is a missed relationship gap between quality and MCH. Quality doesn't know about works performed by MCH. Even MCH didn't go to the review meeting. They did not call the MCH coordinators at the meeting. Therefore, MCH should be parallel. Both units should be integrated and work for hand in hand."

In both Bangladesh and Ethiopia, substantial funding resources for QCN and QI initiatives largely came from global partners, including UNICEF and USAID—all prominent global QCN partners. This was particularly evident in Bangladesh, where prior to QCN's emergence, the Government's Quality Improvement Secretariat (QIS) received most of its funding and technical support from international donors, especially UNFPA, UNICEF, and Save the Children. Crucially, these funders were signed on with QCN and involved in regular national meetings with QIS to coordinate efforts.

While respondents in Uganda and Malawi also discussed their long history of QI initiatives, they were not perceived to be synergistic to the establishment of QCN to the same extent as Bangladesh and Ethiopia. In Uganda, respondents highlighted that many of the prior QI initiatives were specifically focused on HIV, including antiretroviral therapy provision and prevention of mother-to-child transmission. For example, in Uganda, the initial 2013 RMNCAH Sharpened Plan for Uganda [73] placed little emphasis on quality. In Malawi, despite general agreement that QCN was built on previous QI efforts, respondents could not specify what those efforts were. Several respondents, especially at the local level, explained how previous MNH projects led by different stakeholders (e.g., UNICEF and Maikhanda Trust) in different districts had introduced QI tools—such as 5S, continuous QI, TQM, mentorship—before the QCN activities were implemented, but that there was no integration and sometimes duplication between the different projects given the lack of a national strategy guiding quality improvement efforts.

*Data and health system structures and capacity.* Like many LMIC contexts [74], all four countries reported health system capacity challenges, especially concerning human resources and infrastructure. These challenges on the one hand posed challenges for in-country QCN establishment given limited capacities, but in the case of Bangladesh further bolstered conviction and efforts around establishing a networked learning approach, especially given limited resources. Respondents in Bangladesh reported clinical standards not being uniformly applied by all providers, insufficient training and decision aids, long patient wait times, improper queue management, inadequate beds available at clinics, and insufficient medical doctors and cleaning personnel, which contributed to long patient waits and unsanitary conditions in clinics. However, despite these challenges, health workforce morale and motivation were relatively high; they believed activities associated with quality of care were essential to overcoming health system shortcomings.

In contrast, health worker motivation and morale were reported by respondents to be low in Ethiopia, Uganda and Malawi. This posed a challenge for QCN's establishment, given that the establishment of new structures and processes, as well as the focus on quality of care, were perceived to be burdensome and not worth the extra effort, especially in a context where workers already felt overextended. Despite the Ethiopian government's massive scale up of training and education for the health workforce in 2015 [70], there remained a shortage of key maternal and newborn dedicated staff [75], which respondents noted exacerbated low workforce

motivation and high rates of turnover and attrition [76, 77]. This hindered QCN's emergence at the sub-national level. A respondent working at the local level in Ethiopia noted:

> "Due to the change of new laws called [business process reengineering] BPR, the health workers are not paid duty payments and this is also another challenge for workers and administration. Due to lack of incentive, the worker's commitment sharply dropped. They have already lost hope."

In Uganda, respondents described how difficulties with water, electricity, adequate space, roads, and waste management not only made some quality-of-care standards difficult to fulfill, but also contributed to a lack of motivation around QCN's establishment among healthcare workers who were told of the changes they are supposed to make without the capacity to make them. One Ugandan respondent working in an implementing partner organization noted:

> "We cannot really do much [on quality of care] . . ..We do not have that foundation; it is really a pillar in improving processes of small labor units yet the numbers are increasing every year. . .so you are telling a health worker to ensure privacy but this is delivery unit where provision was not made for separating walls between the separating beds."

Respondents in Malawi noted similar reasons contributing to low workforce motivation and commitment and its impact on QCN emergence in their country. One Ministry of Health respondent noted:

> "Motivation of staff [is] an issue. . .. There are a lot of demotivators. . .. frustrations come with small issues like infrastructure in which the staff are working in, career progression, [and lack of government] incentives. Government incentives because some of them are working in remote [areas] but there are no incentives for them."

There were also variances across the countries in terms of data systems for monitoring and evaluation. In Bangladesh, there was some foundational work on a national data system, which helped measure QCN progress. For example, Save the Children worked in MaMoni districts to implement an electronic health record that tracked individual patients by name and personal ID number, and collected general quality improvement information such as stillbirths, deliveries, ANC visits, and family planning services. UNICEF, Save the Children, and QIS worked in the early days of QCN to include QCN indicators in the formal chain of quality data reporting. Furthermore, Management Information Systems in Bangladesh operated through the Directorate General of Hospitals using the widespread DHIS2 software. National level respondents in Bangladesh also noted the existing accountability mechanisms at QCN facilities for data collection and reporting. When data is reported upwards from a local site, it is checked for completeness and accuracy at several levels—first with a district hospital statistician, and then occasionally by quality improvement committee, and then by the Residential Medical Officer, Upazilla Health and Family Planning Officer or by an implementing partner. Ethiopia was also relatively far along in collecting and capturing quality improvement indicators. However, at baseline, the Health Management Information System indicators designated to be used to benchmark improvement were found to be of poor quality, inflated or non-existent [78]. In response, a parallel reporting system for QCN, which was meant to be integrated in the next DHIS revision in 2022, was established. National-level respondents noted that QCN's newly introduced indicators did not align with the previous reporting system, and this jeopardized the feasibility of high-quality data capture:

"The parallel reporting system. . .is our biggest challenge. This is not included in DHIS 2. Extracting these 15 common core indicators from the chart is a big challenge."

In Malawi, parallel reporting systems for experiences of care indicators were also reported. Another challenge related to documentation in Malawi was the poor quality of data collected and poor documentation practices and knowledge of them among health workers. A respondent in the Ministry of Health noted how this made it difficult to assess the outcomes of planned activities, monitor progress and support learning and the scaling up of evidence-based interventions—essential aims of QCN and a critical requirement for QCN emergence. The most recent National Evaluation Platform (NEP) data quality assessment identified material shortages, transportation challenges, limited training opportunities and system level issues (unreliable power and/or internet connectivity, delayed reporting, missing and incomplete reports) as some barriers to data quality [79]. In Uganda, respondents reported monitoring and evaluation to be a weakness given the lack of a clear framework at the national level. Furthermore, the data collected lacked community engagement, with the current strategy not formalizing opportunities to hear from patients and communities at the facility or national level. Respondents had little awareness of ongoing monitoring and evaluation of existing activities or understanding of what happens to the data that has been collected. By not having accurate data to measure potential progress of the network's primary goals (improving mortality rates and user experience) there was less urgency among in-country actors to devote efforts in establishing the network.

*Political developments*. Country leadership commitment and prioritization of MNH and QI, as well broader developments within each of the countries, also shaped the emergence of QCN. Across the four countries, QCN's emergence was supported by the network's alignment with national priorities to improve quality of care, as reflected in national health sector plans, as well as government efforts to make progress on the Sustainable Development Goals (SDGs) [80–83]. A respondent from UNICEF described how these high-level government commitments were critical to QCN emergence in Bangladesh:

"Government has recognized and prioritized that quality is a must. It was access alone before but now [it is understood that] access alone can't reduce preventable death. . .So there is no alternative to quality of care to achieve the sustainable development goals by 2030. The Government understands this very well and they are also committed to this."

In Bangladesh, QCN's emergence was also supported by the Prime Minister's signature in a charter of the United Nations in 2014 and prioritization of community-level care. In Ethiopia, strong alignment between the country's health sector strategic plan and the goal of QCN led the country to be seen as a QCN flagship.

Despite these high-level commitments, there were developments that posed challenges for QCN's emergence. In Ethiopia, the political reality remained that the health system struggled with low healthcare funding and high out-of-pocket expenditure despite the implementation of several reforms in health care financing. Further, respondents in Ethiopia described how the country's historic political transition that began in 2018, as well as the displacement and war in the country, which began in 2020—during the nascent period of QCN—hindered the network's establishment given barriers to freely move, coordinate, and implement the project in affected districts. In Uganda, respondents at the facility level highlighted the negative influence of corruption on healthcare worker morale and the emergence of QCN. They described how money never trickled down to them while working on the frontline, leading to reduced dedication to work and absenteeism—both obstructive characteristics for establishing a

learning and implementation network around quality improvement. In Malawi, respondents described an inability to prioritize QCN given inadequate funding and frequent budget cuts to the health sector. A respondent from a non-governmental organization described how this was not conducive to QCN emergence in the country:

> "There's so many constraints. . .A lot of things have to change and at the moment the structure doesn't allow for that and the political will also doesn't allow for that because every year you find budgets are being cut to the district so how do you expect a district to actually improve services and yet politically you are not committing to that?"

**Network features.** Factors concerning a country's policy environment helped explain the extent and speed of QCN's crystallization across various countries. However, features of the network itself—specifically, its governance structure and leadership—were also just as crucial in explaining QCN's emergence across countries.

*Governance.* QCN crystalized faster and with greater quality when QCN was embedded in powerful national government agencies with previous experience directing QI initiatives. In Bangladesh, the strategic placement and integration of QCN activities into Bangladesh's QIS—a strong and well-established government agency at the center of the country's quality improvement efforts—helped accelerate the network's emergence. Since 2015, QIS strengthened and coordinated quality improvement activities in the public and private health sectors across the country; was responsible for setting quality standards and introducing QI improvement procedures within medical facilities; and was involved in the development of key national quality plans such as the National Quality Strategy for the health sector in 2015, and the fourth National Health Sector Plan (Health Population and Nutrition Sector Program, 2017–2022) [84]. QCN's legitimacy was bolstered early on by having the network's leadership embedded in QIS, which was housed within the Health Economics Unit of the Ministry of Health and Family Welfare (MOHFW) [85, 86]. QIS was initially briefly moved into the Directorate General of Health Services (DGHS), before eventually returning to the MOHFW. Respondents reported this move to be instrumental for QCN's formation, given DGHS's ability to harmonize implementation and monitoring under a single roof. A respondent working at WHO in Bangladesh noted:

> "Quality Secretariat has been shifted to the DGHS. DGHS has another unit of quality assurance. The [previous] gap that existed got reduced. Now quality secretariat and quality assurance are working in the same section. As it is in DGHS, so it's become easier for development partners also to work with them all together directly."

QCN was also embedded in the high-level decision workflows of implementing agencies that had a long history working with government actors in Bangladesh. Specifically, pre-existing relationships between WHO, UNICEF, and Save the Children with government actors were critical to QCN's early formation and success. Each of these actors were simultaneously engaged in QCN and engaged in in-country quality improvement programming. This dual participation led to harmonization between network goals and existing national quality goals. Also, these international agencies and implementing partners had significant reach in the country given pre-existing projects. For example, in the early stages of QCN emergence in Bangladesh, UNICEF had a highly integrated, comprehensive quality improvement approach that covered 119 facilities in 15 districts; the MaMoni MNCSP project worked in 17 districts, all of which have at least some degree of quality improvement programming; and MaMoni/

Save the Children worked in 179 facilities across 14 districts where MNCH quality improvement bundles were being implemented. Furthermore, QCN in Bangladesh purposefully kept its membership open, actively recruiting all major quality-involved actors in the country early on, including government, professional bodies, district health officials, and health facility administrators. Engagement with motivated and engaged frontline health workers strengthened the network's foundations and enabled its intended outputs to diffuse through the Bangladesh national health care system.

In Ethiopia, QCN was also embedded in a strong national government agency: the Federal Ministry of Health of Ethiopia, which played a significant role in setting goals, developing a roadmap and guidelines, and facilitating the implementation of the network in the selected facilities, with the support of implementing partners. Like Bangladesh, there was a strong quality management structure in Ethiopia that had existed since 2015, which helped foster QCN's emergence. Coordination, recognition, commitment, and supervision of QCN was reported by respondents to be strong at the national level, especially as compared to the regional and local levels. Nonetheless, there were clear efforts to adapt and include the regional levels in the early days of the network. A national-level respondent from Ethiopia noted:

> "The national roadmap has undergone regional adaptation, particularly where the WHO regional technical advisor was located, for example, in Oromia, Addis Ababa, Amhara, Dire Dawa, and Harar regions. By doing so, they have also included it in the regional operational plan. Therefore, under the leadership of the Ministry of Health, these partners follow a harmonized and similar approach to implement the project."

However, unlike Bangladesh, QCN in Ethiopia was initially perceived to be a WHO initiative, which hindered the network's initial expansion. One national-level respondent explained this impact on QCN formation:

> "During the initial time, some partners thought this project was WHO's project and hesitated to engage."

These initial perceptions also impacted QCN's formation at the sub-national level, where local respondents reported that their knowledge about the network was very limited and that they perceived it to be a "*two-month campaign activity*". In Ethiopia, QCN formation was largely influenced by implementing partners—such as IHI, CHAI, WHO, Transform HDR, and Transform PHCU—given that the Ministry of Health's facility selection was largely based on their existing activities. Some respondents however noted some challenges with the facility selection, namely: those that were selected did not have critical problems, and the ongoing conflict limited the geographic reach of the network. A regional respondent described how the selection was not representative of the country:

> "The number of facilities is high but does not represent the country as a pilot study. Attempts to include both agrarian & pastoralist areas were made. I don't think they are representative. Some regions were excluded, Somali and Dire Dawa. It is difficult to say that 48 out of 4,000 is representative. The selection criteria were not clear and I think it followed the donor's interest."

In contrast, QCN in Uganda was initially embedded in and co-led by the government's Quality Assurance Department and MCH department. This initially caused fragmentation in activities, given the different mandates and visions of the two institutional leaders tasked to

oversee QCN activities. This resulted in development partners filling the void in the initial years, and little proactive QCN development. For example, the majority of involved partners requested to join the network themselves, rather than being invited by the government, with exception to CHAI and Makerere University School of Public Health. Those initially engaged in QCN were organizations or partners that were already Ministry partners on MNH issues, which respondents noted as a weakness, given the lack of engagement with other external actors, such as Village Health Teams (VHTs), parliamentarians, religious leaders, private providers, and media. One respondent working at WHO noted a lack of community engagement in QCN's early period:

"We want to bring in cultural leaders and mothers to appreciate this whole concept of quality improvement and their role. . . [At] the global level, the community engagement was not defined very well until recently."

Several respondents noted that the lack of community engagement was purposeful; initially, Ministry of Health officials did not believe facilities needed to be aware of the QCN as they were not participating in higher-level network meetings. A government respondent explained:

"I don't think facilities were meant to be part of the Network. They actually don't even attend those quarterly network meetings when we have quarterly calls with WHO. I think we should have those people as members of the Network so that they are consciously aware that they are part of the Global Network."

QCN's slow emergence in Uganda was also partly attributed to the siloed and disjointed nature of interactions among implementing partners in the country. Despite the WHO convening a WhatsApp group for MNH with the UN agencies in Uganda, including UNFPA and UNICEF, many of these agencies worked independently, given different mandates, regions of focus, and use of different tools. While there were attempts to have these partners pool their resources at the national level for QCN activities, respondents reported partners resisted this with reported suspicion around how these resources will be utilized by the Ministry of Health. Consequently, funding and implementation of activities at the national level was initially largely divided by partner, with different partners overseeing various numbers of districts across the country. Respondents described the allocations to be based on previous or ongoing partner work, rather than being intentionally assigned or coordinated. Furthermore, partners were given a high level of autonomy in terms of where, how and what they decided to implement. This lack of clarity and coordination among key network actors, in addition to a lack of trust among those involved, hampered the network's emergence, as well as its subsequent effectiveness [34]. Respondents in Uganda shared that its lack of communication with the global level, outside of few individuals within the Ministry of Health and WHO in Kampala, was partly driven by the perception that the country was being judged against the other network countries without sufficient support to succeed, despite having a shared desire to do so. Similarly, there was a lack of communication with the facility level, which had little awareness of QCN's existence. A government respondent explained:

"When it comes to the facilities, we don't even talk about Networks. . .It is the national level partners that are members of the Network, so the facilities might not be aware when one mentions that they are one of the 10 countries participating in this global Network for MNCH Quality of Care."

It was not until 2019, when the Department of Quality Assurance was renamed to the Standards Compliance Accreditation Patient Protection (SCAPP) under the Directorate of Governance and Regulation, and it was assigned sole oversight and appointed a focal person for QCN, that the network began to emerge in Uganda. This new arrangement enabled SCAPP to bring more partners and funding on board, and work better with other departments in coordinating and implementing QCN activities.

In Malawi, the QCN was integrated into the Quality Management Directorate (QMD), which was established in the Ministry of Health and Population in November 2016 to provide strategic leadership and coordinate quality management and improvement initiatives cross the health sector [87]. Just prior to the launch of QCN, QMD had established quality management structures within health facilities to improve quality of care. These structures, which included quality management focal persons (e.g., district and facility level), a technical working group, quality improvement support teams, and work improvement support teams, were likely to help facilitate QCN's eventual emergence in Malawi. A government respondent noted:

"They [QCN] are using the structures which the ministry or the quality management department has laid to guide the quality of care as a country. . .We don't have like specific structures for quality-of-care network; we use the existing structures from the ministry through the quality management department."

Despite these facilitators, several governance factors contributed to QCN's slow and fragmented emergence in Malawi. First, the planning, development, and dissemination of strategic documents at the national level was slow, as reported by multiple involved respondents. For example, the formulation of strategic documents, including the QCN roadmap and adoption of quality-of-care standards took almost two years to complete. Learning facilities described receiving the standards and other important strategic documents, including the terms of reference for implementing structures (e.g., QIST, WITs, and Ombudsmen) around early- to mid-2019, leaving QI teams unaware of their roles and responsibilities prior to this. Second, there were irregular QCN coordinating meetings and even periods of complete inactivity. Several respondents working at the local level noted this:

"Sometimes meetings are very sporadic but we need to meet regularly and discuss."

"We are supposed to be meeting quarterly but as I have told you it's been a long time."

This contributed to QCN members not having clear direction, as reported by a local level respondent:

"For the first two years of the quality-of-care network, mostly there was no clear direction about what is supposed to be achieved. . . We were aware of the goal that is before us, but in terms of [now saying] 'let us plan, these are the activities that should happen,' one, and two; 'what are the targets that we are should give for each specific learning site?' and then 'what are the activities that we can do to implement it?' So, I think we hadn't made a lot of progress."

Finally, like the other countries, there was disproportionate attention on QCN structures at the national level, as compared to the district and local levels.

*Leadership*. Effective leadership was also crucial to bringing relevant actors across QCN together and in guiding them to effective collective action. In Bangladesh, several respondents working in multiple UN agencies highlighted the effective leadership of the former DGHS

Director for Hospitals and QIS point person, as a crucial driver of a variety of quality activities, including QCN's formation and initial success:

> "He is the driving force. He has brought it [QCN] forward."

Respondents also noted another well-respected champion that was crucial to QCN emergence, given his long history of improving healthcare in Bangladesh. Specifically, this champion implemented the Chowgacha to Jhenaidah Model (CJ Model), which led to improvement of hospital services and resources for underserved people. Between 1996 and 2012, upazila level standard services were developed under his leadership in the Chowgacha Health Complex of Jessore (southwestern district of Bangladesh) and 46 additional workers from the community were mobilised. The Chowgacha Health complex was subsequently awarded best performance in emergency obstetric care in the administrative division consecutively between 2005 and 2014. One respondent working at the national level noted the significance of his efforts at both national and global levels:

> "The initiatives taken by [this particular champion] are not only nationally, but also internationally appreciated. The foreign delegate of UNICEF went [to see the Model]. [It was] also [recognized] nationwide [by the] Prime Minister."

In contrast, there was initially no clear focal person in Ethiopia, as noted by several national-level respondents, which may have slowed QCN's emergence in the country:

> "Since the directorate was not strengthened and there was no MNH quality of care focal person in the directorate, the technical assistant was performing those tasks. Somehow it was good but this created some gap."

There was also little leadership identified in Uganda and Malawi. In Uganda, leadership capacity largely varied across the six districts that were selected to participate, with respondents noting that leadership at these levels were not well empowered or supportive. In Malawi, respondents especially highlighted leadership difficulties at lower levels, with individuals in leadership positions failing to lead by example and unclear leadership structures. One local-level respondent noted:

> "Another challenge that we are facing is leadership. When I say leadership, I don't mean the DMHD and the other topmost leaders because they know why they are there. . . but within our teams, within our health facilities. . .You cannot differentiate who is on top and who is subordinate. . . So all these are the things we consider simple are the things that are highly contributing into negatively results."

## Discussion

Across the four countries examined, there were multiple factors that posed significant challenges to QCN emergence. These included a lack of skilled professionals, inadequate infrastructure, and lacking or unsustainable funding for quality of care and MNH initiatives and programs. However, all four countries also possessed favorable dynamics for QCN emergence —the reason they were selected as pathfinding countries by WHO and global implementing partners. These included: existing programs and policies that advanced quality of care and

MNH improvement, political commitment to SDG3 and universal health coverage, and in-country presence of non-governmental partners also part of QCN global leadership. Nonetheless, QCN emerged more rapidly and to a greater extent in Bangladesh, followed by Ethiopia, Uganda, then Malawi.

Factors connected to the policy environment and network features explained variance in network emergence. With respect to the policy environment, pre-existing resources and initiatives dedicated to MNH and QI, strong data and health system capacity and alignment, and national commitment and advancement on synergistic goals—beyond the primary aims of the network—were crucial drivers to QCN's emergence. With respect to the features of the network itself, the embedding of QCN leadership in powerful agencies with established coordination structures and trusting relationships with key stakeholders, as well as effective leadership —at various levels—were also crucial in explaining the strength of QCN's emergence across countries.

Three strategic considerations are gleaned from this analysis for improving network establishment. The first concerns the nature of the relationships between global and national stakeholders before and during the early days of establishing a network. It is especially critical for global actors to have pre-existing and trusting relationships with national and local leaders [88–90]. This highlights a deeper point: pathfinder countries for QCN—like many global health and development initiatives—are purposefully selected by global development actors given their perceived favorable environment and the partner's existing relationships and presence in-country. These countries, the so-called "aid darlings" [91], which attract the investments of a wide variety of funders and development partners, are naturally more likely to produce cohesive and strong mechanisms over time that facilitate effective coordination and strengthen existing relationships among in-country and foreign partners. This observation presents an inherent tension for global health development partners: pursue policy environments that are most suitable to have the best odds of getting the network off the ground or pursue contexts with the most major challenges that are most in need of the network. The former is more likely to increase inequality; the latter is more likely to fail completely.

Second, networks are more likely to emerge when country actors are engaged early on and they feel invested in the work of the proposed network. This is especially the case when considering the engagement and morale of street level bureaucrats—"the public service workers that interact directly with citizens in the course of their jobs and who have substantial discretion in the execution of their work." [92] Networks crystalize more robustly when these local actors exhibit high levels of morale and motivation and they are made aware of their roles and potential impacts early on. This occurred clearly with QCN in Bangladesh, where local stakeholders —particularly local healthcare providers—perceived themselves as pioneers of the network, rather than as only beneficiaries of network activities, or perceiving activities as burdensome. Strong established systems of governance and data collection were critical to fostering this engagement between stakeholders at the local, national and global levels, which provided mechanisms for coordination and communication, accountability, and continuity between actors and programs.

Third, network emergence requires strong focal leadership, both in terms of individuals and institutions, that pre-exist the network. Networks are more likely to robustly emerge in countries where institutional leadership is embedded in a strong governmental agency that has pre-existing experience leading and coordinating related activities. In Bangladesh and Ethiopia, QCN was embedded in a strong and experienced quality improvement management structure that pre-dated QCN and was housed in a relatively strong Ministry. In contrast, Uganda faced considerable difficulty in identifying an appropriate and capable institution in which to embed QCN national activities in its early days. These experiences shaped the nature

of interactions and extent of coordination among involved national- and local-level stakeholders in the early days of the network's establishment. Furthermore, the presence of charismatic and respected in-country champions is especially crucial in bringing actors together and giving the network legitimacy [30]. In Bangladesh, the buy-in and involvement of several nationally recognized quality improvement champions gave QCN immediate credibility within the country, contributing to the relatively fast uptake of network activities.

This study has several key strengths. The first is its comparative design. While there are exceptions [47, 93], most studies examining global health networks or global health partnerships are single case studies that examine the governance, evolution or impacts of these entities at the global level or within one country context. While important, these studies are limited in their generalizability; it is only through a comparison of one network or partnership across multiple countries that one can one gain in-depth insights (internal validity), as well as generalizability beyond the cases examined (external validity). The second strength lies in the rigor and depth of the data collection and analysis. Multiple types of data (including literature, key-informant interviews, and observation) were triangulated to ensure accuracy. Finally, the subject of this study—network emergence—has largely been under-studied, especially as compared to the function, impact and effectiveness of networks in global health [3, 8, 11, 94–98].

The primary limitation of this study concerns difficulty with operationalizing and detecting network emergence, especially given the prospective nature in which the data for this study was collected—shortly after the time that QCN was conceptualized at the global level, and during the time in which these networks were beginning to take shape in pathfinder countries. Also, given the relative dearth of research on network emergence, the theory that we drew on to examine network emergence was largely developed for examining global networks, rather than in-country networks. While we tried to account for this by also drawing on the global health partnership scholarship, there may be national and local-level dynamics that may have been missed in probing the factors shaping network emergence in these contexts. Future research on this subject should expand to examining emergence of QCN in other pathfinder countries retrospectively, building on what has been gleaned from this study. These studies could examine identified network strengths, such as country leadership for the issue, in greater in-depth. For example, what are the characteristics of effective leadership that can foster network emergence? In the same way, future studies should focus on how to overcome identified weaknesses, which obstruct network emergence. For example, what strategies may be utilized to overcome inherent trust deficiencies among state actors at various levels, and external implementing partners and funders?

## Conclusion

The findings are not only relevant for improving QCN emergence and subsequent activities in other countries, but also in making multi-country networks for other issues in global health—especially at the earliest stages—more successful. This is especially critical given the growing role over the last two decades that global health networks and partnerships have played in global health [99]. These include and are not limited to The Partnership for Maternal, New-born, and Child Health, Stop TB, the GAVI Alliance, and the Global Fund to Fight AIDS, Tuberculosis and Malaria. These networks are central to providing technical assistance and capacity support, engaging in advocacy, and providing financing. They also play pivotal roles in making in-country progress on the Sustainable Development Goals and improving health system function in LMICs.

The factors identified—both relating to the policy environment and the nature of the network—not only shape the speed and way networks are established, but also have implications

for the network's perceived legitimacy [30] and the trajectory of the network's later development and effectiveness [34]. By understanding a network's emergence, practitioners may be in a better position to predict future outcomes, and also more effectively strategize to minimize or resolve less than favorable developments at the inception phase, in order to achieve long-term success. In addition to these practical contributions, this analysis adds to the theoretical scholarship on network emergence, which is a significantly smaller body of scholarship as compared to that examining network effectiveness in global health. The analysis highlights that QCN's emergence lies on a spectrum and identifies the crucial role that strategies and actions of members involved in the network, as well as the policy environment that they operate in, played in explaining network emergence variance across four pathfinding countries.

## Supporting information

**S1 Table. Role of key QCN actors in four pathfinding countries.**
(DOCX)

**S1 Text. PLOS global health QCN evaluation collection 2-page summary.** https://doi.org/10.1371/journal.pgph.0001751.s001.
(DOCX)

**S2 Text. QCN papers common methods section.** https://doi.org/10.1371/journal.pgph.0001751.s002.
(DOCX)

**S1 Checklist. Inclusivity in global research.**
(DOCX)

## Acknowledgments

We thank all respondents and stakeholders for their time and contributions toward making this work possible. The QCN Evaluation Group is: Nehla Djellouli, Kasonde Mwaba, Callie Daniels-Howell, Tim Colbourn (UCL Institute for Global Health, UK), Kohenour Akter, Fatama Khatun, Mithun Sarker, Abdul Kuddus, Kishwar Azad (BADAS-PCP Bangladesh), Kondwani Mwandira, Albert Dube, Gladson Monjeza, Rachel Magaleta, Zabvuta Moffolo, Charles Makwenda (Parent and Child Health Initiative, Malawi), Mary Kinney, Fidele Mukinda (independent researchers, South Africa), Mike English (Oxford University), Yusra Shawar, Will Payne, Jeremy Shiffman (Johns Hopkins University, USA), Kathy Lubowa, Agnes Kyamulabi, Hilda Namakula, Gloria Seruwagi (Makerere University, Uganda), Anene Tesfa, Asebe Amenu, Theodros Getachew, Geremew Gonfa (Ethiopia Public Health Institute, Ethiopia), Seblewengel Lemma, Tanya Marchant (LSHTM, UK).

## Author Contributions

**Conceptualization:** Yusra Ribhi Shawar, Nehla Djellouli, Mary Kinney, Gloria Seruwagi, Mike English, Jeremy Shiffman, Tim Colbourn.

**Data curation:** Nehla Djellouli, Kohenour Akter, Mary Kinney, Kasonde Mwaba, Gloria Seruwagi, Jeremy Shiffman, Tim Colbourn.

**Formal analysis:** Yusra Ribhi Shawar, Nehla Djellouli, Kohenour Akter, Will Payne, Mary Kinney, Kasonde Mwaba, Gloria Seruwagi, Tim Colbourn.

**Funding acquisition:** Yusra Ribhi Shawar, Gloria Seruwagi, Mike English, Jeremy Shiffman, Tim Colbourn.

**Investigation:** Yusra Ribhi Shawar, Nehla Djellouli, Kohenour Akter, Will Payne, Mary Kinney, Kasonde Mwaba, Gloria Seruwagi, Mike English, Tanya Marchant, Jeremy Shiffman, Tim Colbourn.

**Methodology:** Yusra Ribhi Shawar, Nehla Djellouli, Kohenour Akter, Will Payne, Mary Kinney, Kasonde Mwaba, Gloria Seruwagi, Jeremy Shiffman, Tim Colbourn.

**Project administration:** Nehla Djellouli, Kohenour Akter, Kasonde Mwaba, Gloria Seruwagi, Tim Colbourn.

**Resources:** Nehla Djellouli, Tim Colbourn.

**Software:** Nehla Djellouli, Tim Colbourn.

**Supervision:** Yusra Ribhi Shawar, Nehla Djellouli, Mike English, Tanya Marchant, Jeremy Shiffman, Tim Colbourn.

**Validation:** Yusra Ribhi Shawar, Nehla Djellouli, Kohenour Akter, Will Payne, Mary Kinney, Kasonde Mwaba, Gloria Seruwagi, Tanya Marchant, Tim Colbourn.

**Visualization:** Yusra Ribhi Shawar, Will Payne, Tim Colbourn.

**Writing – original draft:** Yusra Ribhi Shawar.

**Writing – review & editing:** Yusra Ribhi Shawar, Nehla Djellouli, Kohenour Akter, Mary Kinney, Kasonde Mwaba, Gloria Seruwagi, Mike English, Tanya Marchant, Jeremy Shiffman, Tim Colbourn.

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
