## [Decision Letter · Decision Letter 0]

30 Apr 2024

PGPH-D-23-00519

Factors Shaping Network Emergence: A Cross-Country Comparison of Quality of Care Networks in Bangladesh, Ethiopia, Malawi, and Uganda

Dear Dr. Shawar,

Thank you for submitting your manuscript to PLOS Global Public Health. After careful consideration, we feel that it has merit but does not fully meet PLOS Global Public Health’s publication criteria as it currently stands. Therefore, we invite you to submit a revised version of the manuscript that addresses the points raised during the review process.

We look forward to receiving your revised manuscript.

Kind regards,

Rachel Hall-Clifford

Section Editor

Journal Requirements:

Additional Editor Comments (if provided):

Reviewers' comments:

Reviewer's Responses to Questions

**Comments to the Author**

1. Does this manuscript meet PLOS Global Public Health’s publication criteria? Is the manuscript technically sound, and do the data support the conclusions? The manuscript must describe methodologically and ethically rigorous research with conclusions that are appropriately drawn based on the data presented.

Reviewer #1: Yes

Reviewer #2: Yes

2. Has the statistical analysis been performed appropriately and rigorously?

Reviewer #1: Yes

Reviewer #2: N/A

3. Have the authors made all data underlying the findings in their manuscript fully available (please refer to the Data Availability Statement at the start of the manuscript PDF file)?

Reviewer #1: Yes

Reviewer #2: No

4. Is the manuscript presented in an intelligible fashion and written in standard English?

Reviewer #1: Yes

Reviewer #2: Yes

5. Review Comments to the Author

Reviewer #1: in general, the paper provides rich information to explain why QCN in selected pathfinder countries has emerged differently. I do enjoy reading it even though sometime I get loss due to the paper is so long especially in the result part. if the authors make the result part more concise will be excellent.

Reviewer #2: Thank you for the opportunity to review the article entititled, "Factors Shaping Network Emergence: A Cross-Country Comparison of Quality of Care Networks in Bangladesh, Ethiopia, Malawi, and Uganda" I have followed this network for several years and enjoyed reading this detailed narrative and analysis of the QCN implementation. The authors are to be commended for bringing clear lessons and recommendations for future networks. I only have two minor comments:

1. Figure 1 is very blurry and hard to read with the color variation in the boxes and black font. Consider revising.

2. Page 10, Lines 166 to 174 provides great information on the roles of the different organizations. Can authors speak to funding at the country level? Was there a discussion or expectations by the network of how the work was to be funded at country level? I think readers will want to understand that beyond the funding by BMGF for the QCN secretariat and UNICEF.

6. PLOS authors have the option to publish the peer review history of their article (what does this mean?). If published, this will include your full peer review and any attached files.

**Do you want your identity to be public for this peer review?** For information about this choice, including consent withdrawal, please see our Privacy Policy.

Reviewer #1: **Yes: **samrit Srithamrongsawat

Reviewer #2: No

---

## [Editor Report · Decision Letter 1]

10 Jun 2024

Factors Shaping Network Emergence: A Cross-Country Comparison of Quality of Care Networks in Bangladesh, Ethiopia, Malawi, and Uganda

PGPH-D-23-00519R1

Dear Dr. Shawar,

We are pleased to inform you that your manuscript 'Factors Shaping Network Emergence: A Cross-Country Comparison of Quality of Care Networks in Bangladesh, Ethiopia, Malawi, and Uganda' has been provisionally accepted for publication in PLOS Global Public Health.

Best regards,

Rachel Hall-Clifford

Section Editor